# Under-Five Mortality and Associated Factors: Evidence from the Nepal Demographic and Health Survey (2001–2016)

**DOI:** 10.3390/ijerph16071241

**Published:** 2019-04-08

**Authors:** Pramesh Raj Ghimire, Kingsley E. Agho, Osita Kingsley Ezeh, Andre M. N. Renzaho, Michael Dibley, Camille Raynes-Greenow

**Affiliations:** 1School of Science and Health, Western Sydney University, Locked Bag1797, Penrith, NSW 2571, Australia; K.Agho@westernsydney.edu.au (K.E.A.); ezehosita@yahoo.com (O.K.E.); 2School of Social Sciences and Psychology, Western Sydney University, Locked Bag1797, Penrith, NSW 2751, Australia; andre.renzaho@westernsydney.edu.au; 3Sydney School of Public Health, The University of Sydney, Edward Ford Building (A27), Sydney, NSW 2006, Australia; michael.dibley@sydney.edu.au (M.D.); camille.raynes-greenow@sydney.edu.au (C.R.-G.)

**Keywords:** risk factors, child mortality, infants, mortality rates, Nepal

## Abstract

Child mortality in Nepal has reduced, but the rate is still above the Sustainable Development Goal target of 20 deaths per 1000 live births. This study aimed to identify common factors associated with under-five mortality in Nepal. Survival information of 16,802 most recent singleton live births from the Nepal Demographic and Health Survey for the period (2001–2016) were utilized. Survey-based Cox proportional hazard models were used to examine factors associated with under-five mortality. Multivariable analyses revealed the most common factors associated with mortality across all age subgroups included: mothers who reported previous death of a child [adjusted hazard ratio (aHR) 17.33, 95% confidence interval (CI) 11.44, 26.26 for neonatal; aHR 13.05, 95% CI 7.19, 23.67 for post-neonatal; aHR 15.90, 95% CI 11.38, 22.22 for infant; aHR 16.98, 95% CI 6.19, 46.58 for child; and aHR 15.97, 95% CI 11.64, 21.92 for under-five mortality]; nonuse of tetanus toxoids (TT) vaccinations during pregnancy (aHR 2.28, 95% CI 1.68, 3.09 for neonatal; aHR 1.86, 95% CI 1.24, 2.79 for post-neonatal; aHR 2.44, 95% CI 1.89, 3.15 for infant; aHR 2.93, 95% CI 1.51, 5.69 for child; and aHR 2.39, 95% CI 1.89, 3.01 for under-five mortality); and nonuse of contraceptives among mothers (aHR 1.69, 95% CI 1.21, 2.37 for neonatal; aHR 2.69, 95% CI 1.67, 4.32 for post-neonatal; aHR 2.01, 95% CI 1.53, 2.64 for infant; aHR 2.47, 95% CI 1.30, 4.71 for child; and aHR 2.03, 95% CI 1.57, 2.62 for under-five mortality). Family planning intervention as well as promotion of universal coverage of at least two doses of TT vaccine are essential to help achieve child survival Sustainable Development Goal (SDG) targets of <20 under-five deaths and <12 neonatal deaths per 1000 births by the year 2030.

## 1. Introduction

Globally, approximately 5.6 million deaths of children under-five years of age (under-five deaths) were reported in the year 2016. Of these deaths, three million occurred between 1 and 59 months of age while the remainder occurred between birth and the first month of life [1]. Deaths occurring at these age periods remain a huge public health concern, especially in sub-Saharan Africa (SSA) and South Asia (SA) including Nepal. An overwhelming majority (80%) of the world’s estimated under-five deaths occurred in SSA and SA, and most are preventable or treatable. Preterm birth complications, pneumonia, malaria and diarrhea contribute approximately 16%, 13%, 5% and 8% of these deaths, respectively [1,2]. In 2015, neonatal tetanus accounted for about 10,000 neonatal deaths in South Asia including Nepal [3]. 

A recent Nepal Demographic and Health Survey (NDHS) report revealed that over a 15-year period, Nepal’s under-five mortality rate (U5MR) decreased by approximately 57%, from 91 deaths per 1000 live births in 2001 to 39 in 2016 [4]. However, under-five mortality in Nepal still remains higher than the Sustainable Development Goal (SDG) target of 20 per 1000 live births [4]. Therefore, to achieve the child survival SDG target, substantial efforts are needed mainly in developing countries such as Nepal. 

Previous studies that examined child mortality in Nepal found that the use of antenatal Iron and folic acid (IFA) supplementation; tetanus toxoid (TT) vaccination during pregnancy; lack of skilled birth attendance, lack of antenatal care (ANC) visits; older maternal age; use of polluting fuel; higher parity, mortality inequality among poor household and mothers with no schooling were linked with child mortality [5,6,7,8,9,10,11]. However, these studies were either community-based experimental in smaller settings or community-based or population based cross-sectional studies; and did not restrict their analysis to the most recent singleton live births in order to reduce recall bias. There is evidence to suggest that multiple births are biologically more likely to die during infancy than with singletons [12]. Additionally, studies disaggregating analyses by different age ranges of the first 59 months of life have been limited in Nepal, especially for the post-neonatal and child mortality subgroups.

This study aimed to identify common factors associated with mortality across all age subgroups from 0 to 59 months of life (neonatal: 0–30 days, post-neonatal: 1–11 months, infant: 0–11 months, child: 12–59 months, and under-five: 0–59 months) using survival information of most recent singleton live births from the NDHS data for the years 2001, 2006, 2011, and 2016. Findings obtained will assist health administrators and public health researchers, as well as government policy makers, to re-evaluate and revitalize existing intervention strategies to accelerate the reduction of under-five mortality in Nepal. 

## 2. Materials and Methods 

### 2.1. Data Source

The NDHS data for the year 2001 [13], 2006 [14], 2011 [15] and 2016 [4] were combined to yield a large sample size of reported deaths. Stratified multi-stage cluster sampling design was used to collect NDHS data and the procedures for collecting data were similar across the surveys (2001–2016). The details of survey methods, sampling techniques and questionnaires used in the NDHS surveys have been described elsewhere [4,13,14,15]. A weighted sample of 16,802 singleton most recent live births five years preceding each survey was used for the analysis (2001: *n* = 4714; 2006: *n* = 4029; 2011: *n* = 4118; and 2016: *n* = 3941). In our analyses, 125 multiple births were excluded because of known higher risk of neonatal mortality due to pregnancy complications and preterm birth amongst multiple births compared to singleton births [12,16,17].

### 2.2. Study Outcomes

Study outcomes for this study were derived from reported deaths of under-five children [4,13,14,15], which was disaggregated as neonatal mortality (0–30 days), post-neonatal mortality (1–11 months), infant mortality (0–11 months), child mortality (12–59 months) and under-five mortality (0–59 months). Direct estimates of childhood mortality were calculated using complete maternal birth histories that include date of every live birth (singleton and multiple birth), survival status, current age for living children and age at death of children [4,13,14,15]. 

### 2.3. Covariates

The selection of covariates for this study was based on Mosley and Chen conceptual framework for child survival in developing countries [18], previous studies on child mortality [5,6,7,8,10,11,19], and information available in combined NDHS datasets [4,13,14,15]. Selected covariates variables were categorized into five distinct groups: community level factors, household level factor, individual level factors, environmental factors and health service factors. The community-level factors consist of types of residence (rural or urban) and ecological zone (Terai, Hill and Mountain). The household factor selected was household wealth index which was constructed by using a principle component analysis [20] of the household facilities and assets (electricity, radio, television, bicycle, telephone, and main material of floor) that was common in the four datasets. For the purpose of this study, the household wealth index was divided into three categories. The bottom, 40% of households were arbitrarily referred to as poor households, the next 40% was classified as the middle households and the top 20% was classified as rich households, consistent with previous study [19]. 

The individual-level factors consist of maternal, child and paternal characteristics. Maternal characteristics were religion (Buddhist, Hindu or others), ethnicity (Brahmin/Chettri, Dalit, Janajati or Madhesi), education (secondary/higher, primary or no education), literacy level (can read or cannot read), age (40–49, 30–39, 20–29 or <20), desire for pregnancy (wanted then, wanted later or no more), and occupation (not working, agriculture or skilled/professional). Child characteristics were combined birth rank and birth interval (2nd/3rd birth rank and >2 years’ interval, 1st birth, 2nd/3rd birth rank and ≤2 years’ interval, 4th/higher birth rank and >2 years’ interval or 4th/higher birth rank and ≤2 years’ interval), previous death of a child (no or yes), and child sex (male or female). The only paternal characteristic was education (secondary/higher, primary or no education).

The environmental factors were types of drinking water source, types of sanitation facilities, and types of cooking fuel. We used World Health Organization and the United Nations Children’s Fund Joint Monitoring Program guidelines [21] to construct types of drinking water source and types of sanitation facilities. Types of cooking fuel were categorized as improved (biogas, natural gas, liquefied petroleum gas and electricity) and unimproved (charcoal, wood, coal/lignite, animal dung, kerosene, straw/shrubs/grass, and agricultural crops).

The health service factors were ANC visits (4+ANC visits, 1–3 ANC visits or no ANC visits), TT vaccination during pregnancy (Two or more TT, one TT or no TT), antenatal IFA supplementation (yes or no), place of delivery (health facility or home facility), delivery assistance (doctors/nurses or others), mode of delivery (vaginal or caesarean) and current use of contraceptive at the time of the survey (yes or no).

### 2.4. Statistical Analysis

STATA (version 14.1, Stata Corp, College Station, TX, USA) was used for the study analysis and Survey “SVY” function was employed to adjust for stratified multi-stage cluster sampling procedure. Weighted counts, and percentage of all covariates were first performed. Mortality rates and 95% confidence interval (CI) by year of survey were obtained by using Roja’s approach [22]. In multivariable analysis, survey Cox proportional hazard models were used to examine the independent factors for each of the study outcome. Tobit and truncreg commands in Stata were used to account for censoring and truncation. 

A staged technique [23] was used to determine the final multivariate regression model. In the first stage, year of survey and community level factors (types of residence, and ecological zone) were entered into the baseline model with manual backward elimination process to remove statistically nonsignificant variables (Model 1). In the second stage, household wealth status and individual level factors (religion, ethnicity, mother’s education, father’s education, mother’s literacy level, mother’s age, desire for pregnancy, mother’s occupation, combined birth rank and birth interval, previous death of a child and child sex) were assessed with Model 1 with manual backward elimination process to remove statistically nonsignificant variables (Model 2). This procedure was followed when environmental (types of drinking water source, types of sanitation facilities, and types of cooking fuel), and health service variables (ANC visits, TT vaccination during pregnancy, antenatal IFA supplementation, place of delivery, delivery assistance, mode of delivery and current use of contraceptive) were included in the third (Model 3), and in the fourth (Model 4), respectively. In each stage, the significance level was set at 0.05; and variables that were statistically significant with the study outcomes in the final model (Model 4) were reported in the study. Variables that were statistically significant with the study outcomes in models 1–3 are presented in supplementary tables (Appendix A). Collinearity was tested and reported in the final model.

We also estimated total risk of deaths in each of the sub-age groups in the population between 2001 and 2016 attributable to each of the common significant independent variables across the age groups in the final multivariable model, under the assumption that the association were causal. The adjusted Population Attributable Risk (PAR) was estimated using the formula below, which is similar to that described by Stafford et al. [24].
PAR= Ψ × (aHR-1)/(aHR)(1)
where Ψ is the weighted proportion of deaths during neonatal, post-neonatal, infant, child and under-five period and aHR is the adjusted hazard ratio.

### 2.5. Ethical Considerations

The ethics committees of the ICF International, USA and the Nepal Health Research Council, Kathmandu, approved all surveys. The first author obtained approval from Measure DHS to download and use the data as part of his doctoral dissertation with the School of Science and Health, Western Sydney University, Australia.

## 3. Results

Over the study period, a total of 1474 deaths occurred consisting of 287 (19%) neonatal mortality, 163 (11%) post-neonatal mortality, 450 (31%) infant mortality, 62 (4%) child mortality, and 512 (35%) under-five mortality. A total of 16,290 observations were left-censored, 512 observations were uncensored and 0 observations were right-censored and truncated for under-five mortality.

The majority (79%) of mothers were rural residents, whereas over half (52%) of the study population were from Terai ecological zone (Table 1). Women who received at least two doses of TT vaccine or those who used antenatal IFA were almost equally represented (60% and 61% respectively).

Figure 1 presents the trends in rates of neonatal, post-neonatal, infant, child, and under-five mortality in Nepal. The mortality rates across the five age subgroups were higher among mothers who resided in the mountain ecological zone, who were rural residents, who could not read, who used unimproved sanitation facilities, and those who reported of having a history of previous death of a child (Table 1). It is worthy to note that mortality rates in this study differ sharply from those reported by NDHS because multiple births were excluded, and analysis was restricted to the most recent live births five years prior each survey.

### Factors Associated with Childhood Mortality

Mothers with a history of previous death of a child, who did not receive TT vaccine during pregnancy, or who were not using contraceptives at the time of the survey were significantly associated with neonatal, post-neonatal, infant, child and under-five mortality (Table 2). In order to investigate collinearity in the final model, when TT vaccine was removed and replaced with IFA supplementation; the results indicated that mothers who did not receive IFA supplementation had the higher risk of neonatal [adjusted HR (aHR) 1.49, 95% CI 1.12, 2.00; *p*-value: 0.007], infant (aHR 1.50, 95% CI 1.20, 1.87; *p*-value: <0.001), child (aHR 2.46, 95% CI 2.08, 9.58; *p*-value: <0.001), and under-five mortality (aHR 1.54, 95% CI 1.24, 1.93; *p*-value: <0.001).

During the study period, the estimated proportion of deaths in children attributed to mothers who had had a previous death of a child was 58.2% for neonatal deaths, 51.6% for post-neonatal deaths, 55.9% for infant deaths, 68.6% for child deaths and 57.4% for the overall under-five deaths. Similarly, 23.8%, 50.9%, 39.9%, 35.6% and 24.7% of neonatal, post-neonatal, infant, child and under-five deaths, respectively, were attributed to children whose mothers were not vaccinated for TT during pregnancy (Table 3).

## 4. Discussion

In Nepal, rates of neonatal, post-neonatal, infant, child and under-five mortality have declined over the past 15 years. In this study, mothers who reported previous death of a child, who did not receive TT vaccines during pregnancy, and nonuse of contraceptives among mothers were found to be associated with neonatal, post-neonatal, infant, child or under-five mortality. We also found that mothers aged <20 years, and those who reported of having a first birth were significantly associated with neonatal, post- neonatal, infant and under-five mortality. In addition, mothers who did not use antenatal IFA supplementation were at greater risk of having neonatal, infant, child and under-five mortality.

Despite substantial improvements in reducing overall under-five mortality within the Asia Pacific region, the progress made during MDG period has been uneven across countries [1], and in a country like Nepal, the current child mortality rate stands well above child survival SDG targets (20 under-five mortality and 12 neonatal mortality per 1000 births by the year 2030). 

Our study identified that rates of neonatal, post-neonatal, infant, child and under-five mortality were significantly associated with mothers who reported of having previous death of a child compared to mothers whose previous child survived. This finding is consistent with a previous population-based study conducted in Bangladesh [19]. A plausible reason for this higher risk of mortality may be attributed due to long-term psychological effect of child death on parents [25] resulting in poor nutrition and inadequate essential healthcare given to the surviving children.

The global burden of disease study conducted in 2015 estimated that the neonatal tetanus mortality rate per 100,000 persons in Nepal (778.52) was higher than Bangladesh (442.94), India (314.21), and Pakistan (358.50) [3]. This study found that mothers who did not receive at least two doses of TT vaccines during pregnancy were more likely to report child mortality across all five age subgroups. This finding is consistent with previous population-based studies in Nepal and Bangladesh [8,19]. Blencowe et al. argued that if mothers received the two recommended dosages of TT vaccination during pregnancy, tetanus related under-five mortality would reduce by 94%, particularly during the neonatal period (0–30 days) [26].

In Nepal, 26% of married women of reproductive age have an unmet need for family planning [4,13,14,15]. Our study found that mothers who were nonusers of contraception were significantly more likely to have a child death in all age subgroups. Shah et al. suggested that the use of contraceptives would reduce child mortality by creating a long birth interval [27] and the impact of short birth intervals on child survival has been well documented in previous studies [5,19].

Higher risk of neonatal, post-neonatal, infant and under-five mortality with mothers aged < 20 years compared to older mothers in our study is in agreement with previous studies [19,28]. An increased risk of mortality observed in the current study may be attributed to inadequate use of obstetric and or antenatal care by younger mothers [29], which often leads to preterm births and low birth weights [30]. Additionally, younger mothers are more likely to be poor, uneducated and unemployed [31], which may affect the health of their infants. 

Our study found that first order births had a significantly greater risk of neonatal, post neonatal, infant and under-five mortality compared with subsequent infants. Existing research has argued that an increased risk of mortality among first-born children may be linked to the high number of young women <20 years of age, in particular, having first births [32]. The increased risk of mortality among the first-born children in this study may be attributed to the fact that a substantial proportion (18%) of Nepalese women gave birth before they reached 20 years of age [4,13,14,15]. A conceivable explanation to this may be due to younger mother’s physical and reproductive immaturity as well as poor nutritional intake during pregnancy, which often leads to low birth weight. More importantly, young mothers are infrequent users of maternal health care services [33].

Similar to the findings from studies conducted in Nepal and Indonesia [5,7,34], this study showed that neonatal, infant, child, and under-five mortality were significantly higher among mothers who did not use antenatal IFA supplementation. Literature suggests that iron deficiency is the most common cause of maternal anaemia; and anaemia during pregnancy is associated with higher risk of prematurity and low birth weight [35,36]. The protective effect of antenatal IFA against child mortality may be aggravated due to the fact that IFA help reduce anaemia during pregnancy so as the prematurity and low birth weight. 

The nonsignificant mortality decline in the post-neonatal group in this study may be due to the focus on neonatal mortality of the IFA intervention in Nepal aimed at reducing neonatal mortality. The IFA program in Nepal was expanded across all districts by 2012, which may have resulted a significant reduction of neonatal mortality in 2016 compared with 2011 [37].

### Strengths and Limitations 

This study has several strengths and limitations. First, the study has a great statistical power to detect statistical differences because four NDHS datasets that lies within MDGs period were combined; and the findings can shed light for effective intervention to help achieve SDG child survival targets. Second, the most recent singleton live births five years prior to each survey were considered for analyses to reduce maternal recall bias [5,34]. Third, the data used in this study was nationally representative with average response rate of 97%; therefore, the findings from this study can be generalizable to the entire Nepalese population. Despite these strengths, this study also has some limitations. First, we cannot make casual inference with observational data such as cross-sectional data used in this study. Second, it is also possible that the number of deaths may have been under-reported because only surviving mothers gave an account of their child’s birth and death during the surveys; and hence, the mortality estimates reported in this study may have been under estimated or overestimated. Third, information on the medical history of the child and mother as well as the cause of child death was unknown as verbal autopsy was not conducted in 2001 and 2011 NDHS. Fourth, information on respiratory infections, diarrhea, nutritional and vaccination status were only collected for surviving children and we were not able to include these important variables in this study. 

## 5. Conclusions

We found that mothers with a previous death of a child, who did not receive TT vaccines during pregnancy, and those who were nonusers of contraceptives were at greater risk of having neonatal, post-neonatal, infant, child and under-five mortality in Nepal. Hence, to achieve child survival SDG targets, our findings indicate the need for community-based family planning interventions such as the promotion of contraceptives as well as universal coverage of two recommended doses of TT vaccines during pregnancy, and these interventions should target women from socioeconomically marginalized groups as well as those who have had previous death of a child.

## Figures and Tables

**Figure 1 ijerph-16-01241-f001:**
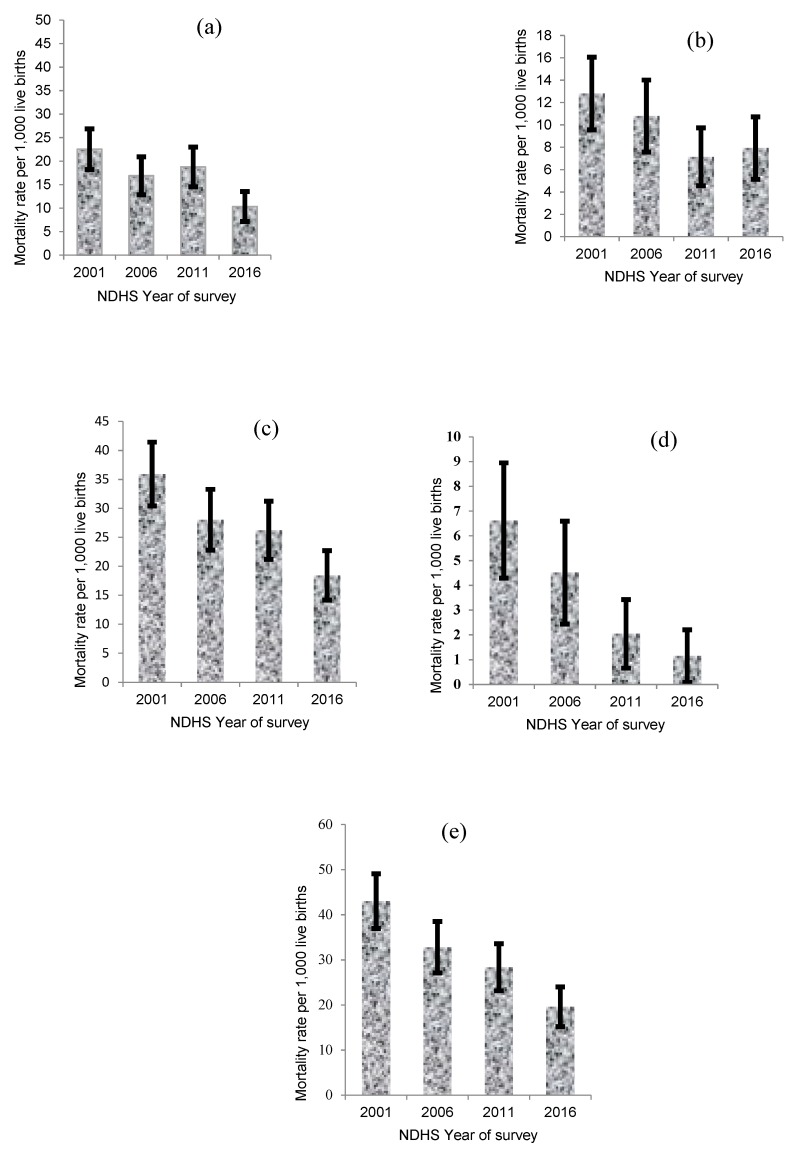
(**a**) Neonatal, (**b**) post-neonatal, (**c**) infant, (**d**) child and (**e**) under-five mortality per 1000 live births with 95% CI by year of survey, Nepal.

**Table 1 ijerph-16-01241-t001:** Characteristics of the population in Nepal for the 2001–2016 waves (*n* = 16,802).

Study Variable	*n* (% *)	NMR ^a^ (95% CI ^f^)	PNMR ^b^ (95% CI)	IMR ^c^ (95% CI)	CMR ^d^ (95% CI)	U5MR ^e^ (95% CI)
Year of survey						
2001	4714 (28)	22 (20, 23)	13 (12, 14)	35 (32, 37)	7 (5, 8)	42 (37, 45)
2006	4029 (24)	17 (15, 19)	11 (9, 12)	28 (24, 31)	4 (3, 5)	32 (27, 36)
2011	4118 (25)	18 (16, 21)	7 (6, 8)	25 (22, 29)	2 (1, 3)	27 (23, 32)
2016	3941 (23)	10 (9, 12)	8 (7, 9)	18 (16, 21)	1 (1, 2)	19 (17, 23)
Type of residence						
Urban	3461 (21)	9 (8, 11)	9 (7, 10)	18 (15, 21)	2 (1, 3)	20 (16, 24)
Rural	13,341 (79)	19 (17, 21)	10 (8, 11)	29 (25, 32)	4 (3, 5)	33 (28, 37)
Ecological zone						
Terai	8663 (52)	18 (16, 20)	9 (8, 11)	27 (24, 31)	4 (3, 5)	31 (27, 36)
Hill	6871 (41)	16 (14, 18)	8 (7, 10)	24 (21, 28)	3 (2, 4)	27 (23, 32)
Mountain	1268 (8)	20 (18, 22)	19 (17, 21)	39 (35, 43)	5 (4, 6)	44 (39, 49)
Wealth index						
Rich	3239 (19)	9 (8, 11)	6 (5, 7)	15 (14, 18)	2 (1, 3)	17 (14, 21)
Middle	6220 (37)	18 (16, 20)	10 (8, 11)	28 (24, 31)	3 (2, 4)	31 (26, 35)
Poor	7343 (44)	20 (18, 22)	11 (9, 13)	31 (27, 35)	5 (4, 6)	36 (31, 41)
Religion						
Buddhist	1191 (7)	14 (12, 16)	5 (4, 6)	19 (16, 22)	4 (3, 5)	23 (19, 27)
Hindu	14,191 (84)	17 (15, 19)	10 (8, 11)	27 (23, 30)	4 (3, 5)	31 (26, 35)
Others	1420 (8)	18 (16, 20)	11 (10, 13)	29 (26, 33)	2 (1, 3)	31 (27, 36)
Ethnicity						
Brahmin/chettri	4735 (28)	15 (13, 16)	9 (8, 11)	24 (21, 27)	2 (1, 3)	26 (22, 30)
Dalit	2550 (15)	22 (20, 24)	10 (9, 12)	32 (29, 36)	5 (4, 6)	37 (33, 42)
Janajati	5694 (34)	12 (11, 14)	8 (7, 10)	20 (18, 24)	4 (3, 5)	24 (21, 29)
Madhesi	3824 (23)	24 (22, 27)	12 (10, 13)	36 (32, 40)	4 (3, 5)	40 (35, 45)
Mother education						
Secondary or higher	5000 (30)	10 (9, 12)	6 (5, 8)	16 (14, 20)	2 (1, 3)	18 (15, 23)
Primary	3010 (18)	14 (12, 16)	11 (10, 13)	25 (22, 29)	2 (1, 3)	27 (23, 32)
No education	8792 (52)	22 (20, 24)	11 (9, 13)	33 (29, 37)	6 (5, 7)	39 (34, 44)
Mother’s literacy level (*n* = 16,800)						
Can read	8828 (53)	12 (10, 13)	8 (6, 9)	20 (16, 22)	2 (1, 3)	22 (17, 25)
Cannot read	7972 (47)	23 (21, 25)	12 (10, 13)	35 (31, 38)	6 (5, 7)	41 (36, 45)
Father’s education (*n* = 16,770)						
Secondary or higher	8647 (51)	14 (12, 15)	9 (7, 10)	23 (19, 24)	3 (2, 4)	26 (21, 28)
Primary	4136 (25)	19 (17, 21)	11 (9, 12)	30 (26, 33)	5 (4, 6)	35 (30, 39)
No education	3986 (24)	23 (21, 25)	11 (9, 13)	34 (30, 38)	5 (4, 6)	39 (35, 44)
Mother occupation (16,800)						
Not working	4226 (25)	15 (13, 17)	6 (5, 7)	21 (18, 24)	2 (1, 2)	22 (20, 24)
Agriculture	10,725 (64)	17 (15, 19)	11 (10, 13)	28 (25, 32)	5 (4, 6)	33 (31, 36)
Skilled/professional	1850 (11)	19 (17, 22)	10 (8, 11)	29 (25, 33)	3 (2, 4)	32 (30, 35)
Mother’s age						
40–49	824 (5)	21 (18, 23)	11 (9, 13)	32 (27, 36)	15 (13, 16)	47 (40, 52)
30–39	4089 (24)	15 (14, 17)	10 (8, 11)	25 (22, 28)	4 (3, 5)	29 (25, 33)
20–29	10,525 (63)	16 (14, 18)	9 (8, 11)	25 (22, 29)	3 (2, 4)	28 (24, 33)
<20	1364 (8)	28 (25, 30)	12 (11, 14)	40 (36, 44)	1 (1, 2)	41 (37, 46)
Mother’s desire for Pregnancy (*n* = 16,801)						
Wanted then	11,695 (70)	18 (16, 20)	10 (9, 12)	28 (25, 32)	4 (3, 5)	32 (28, 37)
Wanted later	2165 (13)	14 (12, 16)	7 (6, 9)	21 (19, 25)	NA	21 (19, 25)
No more	2940 (18)	17 (15, 19)	10 (8, 11)	27 (23, 30)	6 (5, 7)	33 (28, 37)
Birth rank and birth interval						
2nd/3rd birth rank, >2 years	5607 (33)	13 (11, 15)	9 (7, 10)	22 (18, 25)	2 (1, 3)	24 (19, 28)
1st child	4881 (29)	20 (18, 22)	10 (8, 11)	32 (26, 33)	3 (2, 4)	35 (28, 37)
2nd/3rd child, interval ≤2 years	1740 (10)	20 (18, 22)	9 (7, 10)	29 (25, 32)	3 (2, 4)	32 (27, 36)
4th/higher child, interval >2 years	3503 (21)	14 (12, 16)	11 (10, 13)	25 (22, 29)	5 (4, 6)	30 (26, 35)
4th/higher child, interval ≤ 2 years	1072 (6)	31 (28, 33)	11 (10, 13)	42 (38, 46)	11 (10, 13)	53 (48, 59)
Previous Death of a child						
No	13,809 (82)	8 (7, 9)	5 (4, 6)	13 (11, 15)	1 (1, 2)	14 (12, 17)
Yes	2993 (18)	59 (55, 63)	30 (28, 33)	89 (83, 96)	15 (13, 17)	104 (96, 113)
Child Sex						
Male	8822 (53)	17 (15, 19)	8 (7, 10)	25 (22, 29)	4 (3, 5)	29 (25, 34)
Female	7980 (47)	17 (15, 19)	11 (10, 13)	28 (26, 32)	4 (3, 5)	32 (29, 37)
Types of drinking water source (15,659)						
Improved	13,199 (79)	16 (14, 18)	9 (8, 11)	25 (22, 29)	3 (2, 4)	28 (24, 33)
Unimproved	2460 (15)	15 (13, 17)	12 (10, 13)	27 (23, 30)	7 (5, 8)	34 (28, 38)
Types of sanitation facilities (*n* = 15,652)						
Improved	6302 (38)	12 (11, 14)	9 (7, 10)	21 (18, 24)	1 (1, 2)	22 (20, 24)
Unimproved	9350 (56)	19 (16, 21)	10 (9, 12)	29 (25, 33)	5 (4, 6)	34 (29, 39)
Types of Cooking Fuel (15,659)						
Improved	2478 (15)	8 (6, 9)	8 (6, 9)	16 (12, 18)	2 (1, 3)	18 (13, 21)
Unimproved	13,182 (78)	18 (16, 20)	10 (9, 12)	28 (25, 32)	4 (3, 5)	32 (28, 37)
Number of ANC visits (*n* = 16,792)						
4+ANC visits	6660 (40)	9 (8, 11)	8 (6, 9)	17 (14, 20)	2 (1, 3)	19 (15, 23)
1–3 ANC visits	5825 (35)	19 (17, 22)	8 (7, 9)	27 (24, 31)	2 (1, 3)	29 (25, 34)
No ANC visits	4307 (26)	26 (23, 28)	15 (13, 17)	41 (36, 45)	9 (7, 10)	50 (43, 55)
TT Pregnancy Times (*n* = 16,798)						
Two or more TT	10,143 (60)	12 (10, 13)	7 (6, 9)	19 (16, 22)	2 (1, 3)	21 (17, 25)
One TT	2441 (15)	20 (18, 22)	10 (9, 12)	30 (27, 34)	3 (2, 4)	33 (29, 38)
No TT	4214 (25)	29 (26, 31)	15 (13, 17)	44 (39, 48)	8 (6, 9)	52 (45, 57)
IFA supplementation (16,801)						
Yes	10,168 (61)	13 (11, 15)	8 (7, 9)	21 (18, 24)	1 (1, 2)	22 (19, 26)
No	6633 (39)	23 (21, 25)	13 (11, 14)	36 (32, 39)	7 (6, 9)	43 (38, 48)
Place of delivery (16,801)						
Health facility	5462 (33)	12 (11, 14)	8 (7, 10)	20 (18, 24)	1 (1, 2)	21 (19, 26)
Home facility	11,338 (67)	19 (17, 21)	10 (9, 12)	29 (28, 33)	5 (4, 6)	34 (32, 39)
Delivery assistance (16,801)						
Doctors/nurses	4544 (27)	12 (10, 14)	9 (8, 11)	21 (18, 25)	1 (1, 2)	22 (19, 27)
Others	12,257 (73)	19 (17, 21)	10 (8, 11)	29 (25, 32)	5 (4, 6)	34 (29, 38)
Mode of delivery (16,801)						
Non caesarean	16,006 (95)	18 (16, 20)	10 (8, 11)	28 (24, 31)	4 (3, 5)	32 (27, 36)
Caesarean	796 (5)	6 (5, 7)	10 (9, 12)	16 (14, 19)	2 (1, 3)	18 (15, 22)
Current use of contraceptives at the time of the survey						
Yes	6422 (38)	10 (8, 11)	5 (4, 6)	15 (12, 17)	2 (1, 3)	17 (13, 20)
No	10,380 (62)	22 (19, 24)	13 (11, 14)	35 (30, 38)	5 (4, 6)	40 (35, 44)

* Percentage did not add up to 100% because of missing values. ^a^ Neonatal Mortality Rates; ^b^ Post-neonatal Mortality Rates; ^c^ Infant Mortality Rates; ^d^ Child Mortality Rates (CMR); ^e^ Under-five Mortality Rates; ^f^ Confidence Interval.

**Table 2 ijerph-16-01241-t002:** aHR (95% CI) for factors associated with neonatal, post-neonatal, infant, child and under-five mortality in Nepal, 2001–2016, (*n* = 15,750).

Variables	Neonatal Mortality (0–30 Days)		Post-Neonatal Mortality (1–11 Months)		Infant Mortality (0–11 Months)		Child Mortality (12–59 Months)		Under-Five Mortality (0–59 Months)	
	aHR (95% CI)	*p*-value	aHR (95% CI)	*p* -value	aHR (95% CI)	*p* -value	aHR (95% CI)	*p* -value	aHR (95% CI)	*p* -value
Religion										
Buddhist			1.00							
Hindu			2.48 (1.02, 6.03)	0.046						
Others			3.55 (1.27, 9.93)	0.016						
Ethnicity										
Brahmin/chettri					1.00				1.00	
Dalit					1.19 (0.86, 1.65)	0.285			1.16 (0.85, 1.59)	0.341
Janajati					0.84 (0.63, 1.12)	0.239			0.90 (0.70, 1.16)	0.426
Madhesi					1.82 (1.35, 2.45)	<0.001			1.73 (1.29, 2.32)	<0.001
Mother’s literacy level										
Can read	1.00								1.00	
Cannot read	1.57 (1.13, 2.17)	0.007							1.33 (1.03, 1.72)	0.031
Mother’s occupation										
Not working			1.00		1.00				1.00	
Agriculture			1.82 (1.03, 3.22)	0.040	1.45 (1.06, 2.00)	0.022			1.45 (1.06, 1.96)	0.018
Skilled/professional			2.33 (0.99, 5.46)	0.053	2.15 (1.38, 3.35)	0.001			2.15 (1.40, 3.30)	<0.001
Mother’s age										
40–49	1.00		1.00		1.00				1.00	
30–39	1.46 (0.83, 2.58)	0.192	2.20 (1.08, 4.50)	0.031	1.68 (1.07, 2.63)	0.025			1.41 (0.95, 2.08)	0.085
20–29	1.71 (0.97, 3.01)	0.065	3.28 (1.39, 7.77)	0.007	2.09 (1.30, 3.37)	0.002			1.88 (1.24, 2.86)	0.003
<20	2.39 (1.13, 5.05)	0.022	5.04 (1.73, 14.7)	0.003	3.05 (1.64, 5.66)	<0.001			2.76 (1.57, 4.85)	<0.001
Birth rank and birth interval										
2nd/3rd birth rank, >2 years	1.00		1.00		1.00		1.00		1.00	
1st child	2.91 (1.79, 4.74)	<0.001	2.12 (1.10, 4.10)	0.025	2.56 (1.72, 3.80)	<0.001	1.87 (0.54, 6.47)	0.322	2.55 (1.77, 3.68)	<0.001
2nd/3rd birth rank, interval ≤2 years	1.22 (0.75, 1.99)	0.421	1.01 (0.53, 1.92)	0.985	1.15 (0.76, 1.73)	0.501	1.43 (0.44, 4.58)	0.551	1.16 (0.78, 1.73)	0.463
4th/higher birth rank, interval >2 years	0.29 (0.17, 0.50)	<0.001	0.54 (0.27, 1.07)	0.078	0.37 (0.24, 0.56)	<0.001	0.33 (0.14, 0.78)	0.011	0.36 (0.24, 0.52)	<0.001
4th/higher birth rank, interval ≤ 2 years	0.62 (0.34, 1.11)	0.109	0.52 (0.26, 1.05)	0.066	0.60 (0.38, 0.95)	0.028	0.78 (0.39, 1.59)	0.499	0.62 (0.42, 0.91)	0.015
Previous death of a child										
No	1.00		1.00		1.00		1.00		1.00	
Yes	17.33 (11.44, 26.26)	<0.001	13.05 (7.19, 23.67)	<0.001	15.90 (11.38, 22.22)	<0.001	16.98 (6.19, 46.58)	<0.001	15.97 (11.64, 21.92)	<0.001
TT Pregnancy Times										
Two or more TT	1.00		1.00		1.00		1.00		1.00	
One TT	1.65 (1.04, 2.60)	0.033	1.34 (0.78, 2.29)	0.289	1.51 (1.06, 2.16)	0.022	1.62 (0.58, 4.54)	0.353	1.54 (1.09, 2.16)	0.013
No TT	2.28 (1.68, 3.09)	<0.001	1.86 (1.24, 2.79)	0.003	2.44 (1.89, 3.15)	<0.001	2.93 (1.51, 5.69)	0.002	2.39 (1.89, 3.01)	<0.001
Contraceptive use										
yes	1.00		1.00		1.00		1.00		1.00	
No	1.69 (1.21, 2.37)	0.002	2.69 (1.67, 4.32)	<0.001	2.01 (1.53, 2.64)	<0.001	2.47 (1.30, 4.71)	0.005	2.03 (1.57, 2.62)	<0.001

aHR: adjusted Hazard Ratio; Hazard ratio adjusted for: model 3; and number of ANC visits, TT pregnancy times, IFA supplementation, place of delivery, delivery assistance, mode of delivery, and current use of contraceptives at the time of the survey.

**Table 3 ijerph-16-01241-t003:** Estimated PAR with 95% CI for common significant factors for child mortality across the five age groups in Nepal, 2001–2016 (*n* = 15,750).

Variable	Neonatal Mortality			Post-Neonatal Mortality			Infant Mortality			Child Mortality			Under-Five Mortality		
Previous death of a child	*n* *	aHR ^$^	PAR (95% CI)	*n* *	aHR ^$^	PAR (95% CI)	*n* *	aHR ^$^	PAR (95% CI)	*n* *	aHR ^$^	PAR (95% CI)	*n* *	aHR ^$^	PAR (95% CI)
No	38.2	1.00		44.1	1.00		40.4	1.00		27.1	1.00		38.8	1.00	
Yes	61.8	17.3	58.2 (50.5–65.2)	55.9	13.05	51.6 (40.2–62.0)	59.6	15.9	55.9 (49.6–61.7)	72.9	16.98	68.6 (48.7–82.1)	61.2	15.97	57.4 (51.5–62.9)
TT pregnancy times															
Two or more	40.8	1.00		46.8	1.00		43.0	1.00		33.6	1.00		41.8	1.00	
One TT	16.7	1.65	6.58 (0.45–14.2)	15.3	1.34	––	16.2	1.51	5.47 (0.70–11.3)	12.4	1.62	––	15.8	1.54	5.54 (1.01–10.9)
No TT	42.4	2.28	23.8 (14.6–33.2)	37.9	1.86	17.5 (5.81–29.8)	40.8	2.44	24.1 (16.8–31.5)	54.0	2.93	35.6 (14.2–54.5)	42.4	2.39	24.7 (17.7–31.7)
Contraceptive use															
Yes	21.7	1.00		19.0	1.00		20.7	1.00		27.7	1.00		21.0	1.00	
No	78.3	1.69	32.0 (12.5–48.3)	81.0	2.69	50.9 (29.6–66.6)	79.3	2.01	39.9 (25.9–51.6)	77.3	2.47	46.0 (15.1–67.8)	79.0	2.03	40.1 (27.2–51.1)

* Weighted proportion of deaths in each of the five age groups. aHR: adjusted Hazard Ratio; PAR: Population Attributable Risk; CI: Confidence Interval. ^$^ Adjusted model included: model 3; and number of ANC visits, TT pregnancy times, IFA supplementation, place of delivery, delivery assistance, mode of delivery, and current use of contraceptives at the time of the survey. –– PAR was not estimated because variables were not significant.

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
