# Peer review of "Under-Five Mortality and Associated Factors: Evidence from the Nepal Demographic and Health Survey (2001–2016)"

_ijerph, 2019, doi:10.3390/ijerph16071241_

Reviewer 1 Report

The manuscript titled, “Under-5 mortality and associated factors: evidence from the Nepal Demographic and Health Survey (2001-2016)” presents results from a larger sample of youth in Nepal. While some parts of the paper were interesting, other areas could be improved. I hope the authors consider my feedback for enhancing their manuscript.

MAJOR COMMENTS

·         Statistical analysis: have the authors considered adding Kaplan-Meier estimators for their results? While I understand that you can’t control for covariates with these analyses, presenting information for the under 5 cohort especially (because of n= and study period duration) might support the Cox models.

·         Statistical analysis: I think the results for models 1-3 should be reported in an appendix.

·         Statistical analysis: for more detail, please add text for the entry variable, censoring, and truncation.

·         Table 1: given that you are performing Cox regression, it makes sense that you would exclude those with missing covariates from this table (and from your final n=). The models are going to do this for you anyways. Make sure the n= for each exclusion is listed in the text. I think the table would then more accurately depict the analyses. Consider adjusting here and where appropriate in the text.

·         Results: I am having a hard time linking the information that is in Tables 2-3 with what is presented in the text for the results section. For example, results for TT vaccine during pregnancy do not appear to be in Table 2. Please better connect the results from the tables in the text.

·         Abstract: the abstract would benefit from adding specific results (e.g., hazard ratios and 95% CI).

MINOR COMMENTS

·         Line 31: while this reviewer understands what under-5 is, I think adding a sentence that transitions before here, helping to explain what under-5 deaths are, would be beneficial for clarity.

·         Line 33: should be “[1].” Be sure to check for minor typos throughout.

·         Line 40: U5MR has not been defined yet.

·         Line 48: “these studies were either community-based experimental study in” is a plural-singular contradiction. Check for minor grammatical errors throughout.

·         Lines 141-142: please add percentages to match the frequencies.

·         Table 1: the title should just be “Characteristics of the population in Nepal for the 2001-2016 waves.” or similar. All the other information can go under the table as a note. Also, “N=” needs to be added to some of the values in the first column.

·         Line 227: it is not appropriate to introduce results in the Discussion section. Please move this information to the appropriate location in the results section.

·         Line 291: I am not sure if this is considered a cross-sectional design. Maybe just “observational study” instead? You still can’t make causal inferences with observational data.

Author Response

April 5, 2019

The Editor

Subject: Submission of revised manuscript entitled “Under-5 mortality and associated factors: evidence from the Nepal Demographic and Health Survey (2001-2016)” (Ghimire, Agho, Ezeh, Renzaho, Dibley, Raynes-Greenow)”

Dear Editor,

Thank you so much for allowing us to revise and re-submit our manuscript. We are incredibly grateful to the reviewers for insightful feedback that has strengthened our manuscript. We have addressed all the issues raised by the reviewers; and below is point by point response to reviewer’s comments. 

All changes made to the revised manuscript have been marked using track change function in the Microsoft Word.

We hope that the revisions made have properly addressed the reviewers’ concerns, and we believe that our revised manuscript is acceptable for publication in International Journal of Environmental Research and Public Health.

Should you require additional information, please contact me at [email protected]

Once again, on behalf of all the co-authors, thank you so much.

Sincerely Yours

Pramesh Raj Ghimire (Corresponding author)

Reviewer 1

The manuscript titled, “Under-5 mortality and associated factors: evidence from the Nepal Demographic and Health Survey (2001-2016)” presents results from a larger sample of youth in Nepal. While some parts of the paper were interesting, other areas could be improved. I hope the authors consider my feedback for enhancing their manuscript.

Response: Thank you so much for your important feedback.

MAJOR COMMENTS

Comment: Statistical analysis: have the authors considered adding Kaplan-Meier estimators for their results? While I understand that you can’t control for covariates with these analyses, presenting information for the under 5 cohort especially (because of n= and study period duration) might support the Cox models.

Response: To do Kaplan-Meier estimators, we have to compare our estimate by certain covariates (for example compare with mother’s age, antenatal care and so on). But in this study, we are looking at factors associated with childhood mortality across five age sub-groups. However, we produced Kaplan-Meier curve by year of survey for perusal (Please see figure below).

Comment: Statistical analysis: I think the results for models 1-3 should be reported in an appendix.

Response: Thank you for the important suggestion; and we have reported results for models 1-3 (please see supplementary table 1 and supplementary table 2).

Comment: Statistical analysis: for more detail, please add text for the entry variable, censoring, and truncation.

Response: Thank you for the suggestion. We have added text for the entry variables (Please see statistical analysis section, lines 128-138); as well as details for censoring and truncation (Please see line 126-127, and lines 160-161).

Comment: Table 1: given that you are performing Cox regression, it makes sense that you would exclude those with missing covariates from this table (and from your final n=). The models are going to do this for you anyways. Make sure the n= for each exclusion is listed in the text. I think the table would then more accurately depict the analyses. Consider adjusting here and where appropriate in the text.

Response: We agree with the reviewer; and we have included ‘n’ used for the regression models in the table.

Comment: Results: I am having a hard time linking the information that is in Tables 2-3 with what is presented in the text for the results section. For example, results for TT vaccine during pregnancy do not appear to be in Table 2. Please better connect the results from the tables in the text.

Response: Results for TT vaccine during pregnancy has now appeared in table 2 (Please see table 2 of the revised manuscript).

Comment: Abstract: the abstract would benefit from adding specific results (e.g., hazard ratios and 95% CI).

Response: We agree with the reviewer; and we have added hazard ratios and 95% CI for common factors associated with neonatal, post-neonatal, infant, child, and under-5 mortality (Please see abstract section, lines 21-31).

MINOR COMMENTS

Comment: Line 31: while this reviewer understands what under-5 is, I think adding a sentence that transitions before here, helping to explain what under-5 deaths are, would be beneficial for clarity.

Response: For clarity, we have revised our manuscript that explains what under-5 deaths are (Please see introduction section, line 38).

Comment: Line 33: should be “[1].” Be sure to check for minor typos throughout.

Response: Thank you for indicating this, and we have revised the manuscript accordingly (Please see line 40 in the revised manuscript).

Comment: Line 40: U5MR has not been defined yet.

Response: U5MR has been defined in the revised manuscript (Please see introduction section, line 48)

Comment: Line 48: “these studies were either community-based experimental study in” is a plural-singular contradiction. Check for minor grammatical errors throughout.

Response: Thank you for indicating this, and we have corrected grammatical errors throughout.

Comment: Lines 141-142: please add percentages to match the frequencies.

Response: We have added percentages to match the frequencies (Please see lines 158-160 in the revised manuscript)

Comment: Table 1: the title should just be “Characteristics of the population in Nepal for the 2001-2016 waves.” or similar. All the other information can go under the table as a note. Also, “N=” needs to be added to some of the values in the first column.

Response: Thank you for the comment. We have revised the title for table 1 accordingly.

Comment: Line 227: it is not appropriate to introduce results in the Discussion section. Please move this information to the appropriate location in the results section.

Response: Revised as suggested.

Comment: Line 291: I am not sure if this is considered a cross-sectional design. Maybe just “observational study” instead? You still can’t make causal inferences with observational data.

Response: Thanks for the comment. We have revised our manuscript and the text below has been added in the revised manuscript.

“First, we cannot make casual inference with observational data such as cross-sectional data used in this study.” 

Reviewer 2 Report

Mortality of younger children is a very important indicator for assessment of nutritional and health status of younger children, health care service and even the socioeconomic status. Reduction of mortality is a key task around world, especially for developing countries. This study investigated the prevalence of mortality from birth to 5-year of age among Nepal children, and further looked for possible affecting factors. They used larger samples from national survey of Nepal, which increased statistical power. The results provided some implication for improvement of maternal and child health care.

1. When they analyzed the risk factors, the authors pooled the data from 4 surveys of NDHS. How to pool the data? I think they should have the same or similar procedure of collection, which should be mentioned in the section of methods.

2. The variables called as confounding variables by authors, actually, were used for two purposes. One is for estimation of mortality by adjusting for these variables, which could be called confounding variables. Other is to investigate the risk factors. However, authors did not present clearly how to estimate the adjusted mortality rate in the section of methods but just said this question in the discussion. In the current version, the variables in the part of “2.3 Potential confounding variables” seemed to be used for investigation of potential affecting factors by the staged technique. If so, the term of confounding variables might not be appropriate, and covariates is better.

3. the notes of table 2 said, “ Hazard ratio adjusted for: year of survey, types of residence, ecological zone, wealth index, religion, ethnicity, mother’s education, mother’s literacy level, father’s education, mother’s occupation, mother’s age, mother’s desire for pregnancy, birth rank and birth interval, previous death of a child, child sex, types of drinking water source, types of sanitation facilities, types of cooking fuel, number of ANC visits, TT pregnancy times, IFA supplementation, place of delivery, delivery assistance, mode of delivery, Current use of contraceptives at the time of the survey.” Those variables were considered for the whole analysis but not adjusted for last model. Actually, in each model, some variables might be removed if there was not significant statistically. It means that not all of variables are in the final models. So I think that this expression above might be not appropriate. Authors said in the “3.1 Factors associated with ….” that ‘In the final model, when TT vaccine was removed and replaced with IFA supplementation; the results indicated that mothers who did not receive IFA supplementation had the higher risk of neonatal’, what is meaning? What purpose is it? Maybe authors did not present the whole procedure of selection of risk factors but just reported the results from final models.

4. authors just reported the PAR for the same variables for all sub-group of age. How about other significant variable for different age group?

5. The difference in risk factors among 5 sub-group of age in table 2 was clear. Whether this difference induces some special implication of child health care? Some discussion should be required.

Author Response

April 5, 2019

The Editor

Subject: Submission of revised manuscript entitled “Under-5 mortality and associated factors: evidence from the Nepal Demographic and Health Survey (2001-2016)” (Ghimire, Agho, Ezeh, Renzaho, Dibley, Raynes-Greenow)

Dear Editor,

Thank you so much for allowing us to revise and re-submit our manuscript. We are incredibly grateful to the reviewers for insightful feedback that has strengthened our manuscript. We have addressed all the issues raised by the reviewers; and below is point by point response to reviewer’s comments. 

All changes made to the revised manuscript have been marked using track change function in the Microsoft Word.

We hope that the revisions made have properly addressed the reviewers’ concerns, and we believe that our revised manuscript is acceptable for publication in International Journal of Environmental Research and Public Health.

Should you require additional information, please contact me at [email protected]

Once again, on behalf of all the co-authors, thank you so much.

Sincerely Yours

Pramesh Raj Ghimire (Corresponding author)

 Reviewer 2

Mortality of younger children is a very important indicator for assessment of nutritional and health status of younger children, health care service and even the socioeconomic status. Reduction of mortality is a key task around world, especially for developing countries. This study investigated the prevalence of mortality from birth to 5-year of age among Nepal children, and further looked for possible affecting factors. They used larger samples from national survey of Nepal, which increased statistical power. The results provided some implication for improvement of maternal and child health care.

Comment: When they analyzed the risk factors, the authors pooled the data from 4 surveys of NDHS. How to pool the data? I think they should have the same or similar procedure of collection, which should be mentioned in the section of methods.

Response: Thank you for the important suggestion. The manuscript has been revised accordingly to reflect similar procedure of data collection (Please see methods section, line 72).

Comment: The variables called as confounding variables by authors, actually, were used for two purposes. One is for estimation of mortality by adjusting for these variables, which could be called confounding variables. Other is to investigate the risk factors. However, authors did not present clearly how to estimate the adjusted mortality rate in the section of methods but just said this question in the discussion. In the current version, the variables in the part of “2.3 Potential confounding variables” seemed to be used for investigation of potential affecting factors by the staged technique. If so, the term of confounding variables might not be appropriate, and covariates is better.

Response: We agree with the reviewer; and the manuscript has been revised accordingly.

Comment: the notes of table 2 said, “ Hazard ratio adjusted for: year of survey, types of residence, ecological zone, wealth index, religion, ethnicity, mother’s education, mother’s literacy level, father’s education, mother’s occupation, mother’s age, mother’s desire for pregnancy, birth rank and birth interval, previous death of a child, child sex, types of drinking water source, types of sanitation facilities, types of cooking fuel, number of ANC visits, TT pregnancy times, IFA supplementation, place of delivery, delivery assistance, mode of delivery, Current use of contraceptives at the time of the survey.” Those variables were considered for the whole analysis but not adjusted for last model. Actually, in each model, some variables might be removed if there was not significant statistically. It means that not all of variables are in the final models. So I think that this expression above might be not appropriate.

Response: we agree with the reviewer; and to make this more appropriate, we have revised notes for table 2.

Comment: Authors said in the “3.1 Factors associated with ….” that ‘In the final model, when TT vaccine was removed and replaced with IFA supplementation; the results indicated that mothers who did not receive IFA supplementation had the higher risk of neonatal’, what is meaning? What purpose is it? Maybe authors did not present the whole procedure of selection of risk factors but just reported the results from final models.

Response: Thank you for the comment. This meant that two covariates are measuring the same thing and gives a perfect correlation; and hence, In the final model, we tested collinearity by removing and replacing possible covariates one at a time, consistent with a previous study [1].

Comment: authors just reported the PAR for the same variables for all sub-group of age. How about other significant variable for different age group?

Response: the aim of this study was to identify common factors associated with mortality across five age sub-groups (neonatal, post-neonatal, infant, child, and under-5 mortality). Therefore, we reported PAR for those variables which were found to be associated across all age sub-groups.

Comment: The difference in risk factors among 5 sub-group of age in table 2 was clear. Whether this difference induces some special implication of child health care? Some discussion should be required.

Response: The reviewer suggestions are very useful; and we think this is another research paper that can look at suggesting different interventions in order to reduce age specific childhood mortality in Nepal. However, we have provided possible future intervention based on the aim of this paper.

Reference:

Ezeh O, Agho K, Dibley M, Hall J, Page A. The impact of water and sanitation on childhood mortality in Nigeria: evidence from demographic and health surveys, 2003–2013. International Journal of Environmental Research and Public Health. 2014 Sep;11(9):9256-72

Round  2

Reviewer 1 Report

Results: I include the frequency and percentage in the first paragraph (i.e., n (%)).

Author Response

April 4, 2019

The Editor

Subject: Submission of revised manuscript entitled “Under-5 mortality and associated factors: evidence from the Nepal Demographic and Health Survey (2001-2016)” (Ghimire, Agho, Ezeh, Renzaho, Dibley, Raynes-Greenow)”

Dear Editor,

Thank you so much for allowing us to revise and re-submit our manuscript once again.

Any changes in this round of revision have been highlighted using track change function in the Microsoft Word. In addition, below is a point by point response to the reviewer’s concern.

We hope that the revisions made have properly addressed the reviewers’ concerns, and we believe that our revised manuscript is acceptable for publication in International Journal of Environmental Research and Public Health.

Should you require additional information, please contact me at [email protected]

Once again, on behalf of all the co-authors, thank you so much.

Sincerely Yours

Pramesh Raj Ghimire (Corresponding author)

Reviewer 1

Comment: Results: I include the frequency and percentage in the first paragraph (i.e., n (%)).

Response: Thank you for suggestion, and the manuscript has been revised accordingly (please see lines 156-157).

Reviewer 2 Report

The authors have addressed most of my comments, and improve the manuscript much. The following minor problem should be improved further.

Although authors explained about why they test IFA effect instend of TT vaccine. [ “3.1 Factors associated with ….” that ‘In the final model, when TT vaccine was removed and replaced with IFA supplementation; the results indicated that mothers who did not receive IFA supplementation had the higher risk of neonatal’] in order to investigate collinearity. actually, it is not clear at present version. It will better if there are some more statement in the section of statistical analysis.

Author Response

April 4, 2019

The Editor

Subject: Submission of revised manuscript entitled “Under-5 mortality and associated factors: evidence from the Nepal Demographic and Health Survey (2001-2016)” (Ghimire, Agho, Ezeh, Renzaho, Dibley, Raynes-Greenow)”

Dear Editor,

Thank you so much for allowing us to revise and re-submit our manuscript once again.

Any changes in this round of revision have been highlighted using track change function in the Microsoft Word. In addition, below is a point by point response to the reviewer’s concern.

We hope that the revisions made have properly addressed the reviewers’ concerns, and we believe that our revised manuscript is acceptable for publication in International Journal of Environmental Research and Public Health.

Should you require additional information, please contact me at [email protected]

Once again, on behalf of all the co-authors, thank you so much.

Sincerely Yours

Pramesh Raj Ghimire (Corresponding author)

Reviewer 2

Comment: Although authors explained about why they test IFA effect instend of TT vaccine. [ “3.1 Factors associated with ….” that ‘In the final model, when TT vaccine was removed and replaced with IFA supplementation; the results indicated that mothers who did not receive IFA supplementation had the higher risk of neonatal’] in order to investigate collinearity. actually, it is not clear at present version. It will better if there are some more statement in the section of statistical analysis.

Response: Thank you for the comment, and the manuscript has been revised accordingly (please see lines 182-183 of the revised manuscript).